# Effectiveness and safety of acupuncture for postoperative ileus following gastrointestinal surgery: A systematic review and meta-analysis

**Zi Ye**[1☉], **Xuqiang Wei**[1☉], **Shouquan Feng**[2], **Qunhao Gu**[2], **Jing Li**[3], **Le Kuai**[4], **Yue Luo**[4], **Ziqi Xi**[1], **Ke Wang**[1]*, **Jia Zhou**[1]*

1 Acupuncture Anesthesia Clinical Research Institute, Yueyang Hospital of Integrated Traditional Chinese and Western Medicine, Shanghai University of Traditional Chinese Medicine, Shanghai, China, 2 Department of General Surgery, Yueyang Hospital of Integrated Traditional Chinese and Western Medicine, Shanghai University of Traditional Chinese Medicine, Shanghai, China, 3 Department of Acupuncture, Yueyang Hospital of Integrated Traditional Chinese and Western Medicine, Shanghai University of Traditional Chinese Medicine, Shanghai, China, 4 Department of Dermatology, Yueyang Hospital of Integrated Traditional Chinese and Western Medicine, Shanghai University of Traditional Chinese Medicine, Shanghai, China

☉ These authors contributed equally to this work.
* wangke8430@163.com (KW); pdzhoujia@163.com (JZ)

**Data Availability Statement:** All relevant data are within the paper and its Supporting Information files.

## Abstract

### Background

Postoperative ileus (POI) is an important complication of gastrointestinal (GI) surgery. Acupuncture has been increasingly used in treating POI. This study aimed to assess the effectiveness and safety of acupuncture for POI following GI surgery.

### Methods

Seven databases (PubMed, Embase, the Cochrane Library, China National Knowledge Infrastructure, Wan fang Data, VIP Database for Chinese Technical Periodicals, and Chinese Biomedical Literature Database) and related resources were searched from inception to May 30, 2021. Randomized controlled trials (RCTs) reporting the acupuncture for POI in GI were included. The quality of RCTs was assessed by the Cochrane Collaboration Risk of Bias tool, and the certainty of the evidence was evaluated by the Grading of Recommendations, Assessment, Development and Evaluations (GRADE) approach. A meta-analysis was performed by using RevMan 5.4 software.

### Results

Eighteen RCTs involving 1413 participants were included. The meta-analysis showed that acupuncture could reduce the time to first flatus (TFF) (standardized mean difference [SMD] = −1.14, 95% confidence interval [CI]: −1.54 to −0.73, P < 0.00001), time to first defecation (TFD) (SMD = −1.31, 95% CI: −1.88 to −0.74, P < 0.00001), time to bowel sounds recovery (TBSR) (SMD = −1.57, 95% CI: −2.14 to −1.01, P < 0.00001), and length of hospital stay (LOS) (mean difference [MD] = −1.68, 95% CI: −2.55 to −0.80, P = 0.0002) compared with

**Funding:** This work was supported by the 2019 Project of Building Evidence Based Practice Capacity for TCM [2019XZZX-ZJ0011]; the Shanghai Clinical Research Center for Acupuncture and Moxibustion Accelerating [20MC1920500]; the Development of Chinese Medicine Three-Year Action Plan of Shanghai [ZY (2018-2020)-CCCX-2004-04]; the Scientific Research Project of Shanghai Municipal Health Commission [201840011]; the Clinical Key Specialty Construction Foundation of Shanghai [shslczdzk04701]; and the Shanghai Health System Talent Training Program [2018BR24]. The funders had no role in study design, data collection and analysis, decision to publish, or preparation of the manuscript.

**Competing interests:** The authors have declared that no competing interests exist.

usual care. A subgroup analysis found that acupuncture at distal acupoints once daily after surgery had superior effects on reducing TFF and TFD. A sensitivity analysis supported the validity of the finding. Acupuncture also manifested an effect of reducing TFF, TFD and TBSR compared with sham acupuncture but the result was not stable. Relatively few trials have reported whether adverse events have occurred.

## Conclusions

Acupuncture showed a certain effect in reducing POI following GI surgery with very low-to-moderate quality of evidence. The overall safety of acupuncture should be further validated. More high-quality, large-scale, and multicenter original trials are needed in the future.

## Introduction

Postoperative ileus (POI) is one of the most frequently occurring complications of gastrointestinal (GI) surgery that continues to prove challenging [1]. Approximately 24% of patients undergoing colectomy will develop this complication [2]. POI is a pathologic GI tract dysmotility characterized by abdominal distension, pain, delayed passage of flatus and stool, nausea, vomiting, and inability to tolerate an oral diet [3, 4]. POI is also a critical risk factor for severe morbidity, such as dehydration, electrolyte imbalance, or sepsis [5]. These conditions not only decrease the patient's quality of life but also lead to prolonged hospitalization, increased hospital costs, and 30-day readmission rates [6–8]. Consequently, POI imposes a substantial financial and medical resource burden on the healthcare system. The annual costs of POI management in the US have been USD 1.5 billion [9].

The indistinct mechanism and etiology of POI, which involves opioid analgesia, intestinal operation, postoperative stress or anxiolytic medications, increase the difficulty of its prevention and treatment [5, 10]. Effective strategies for POI management to accelerate postoperative GI function recovery is exigent. The usual care patient receives after GI surgery mainly includes routine nasogastric tubes, intravenous fluids, parenteral nutrition, and early mobilization [11]. The POI management adopting multidisciplinary approaches is also recommended, including minimizing surgical manipulation of the intestine, epidural analgesia, and stimulating bowel motility treatments, such as alvimopam, coffee, and chewing gum [12–14]. However, the definite clinical efficacy of those therapies is controversial [15, 16]. There has been a need to seek complementary and alternative medicine approaches for POI management [17].

Acupuncture is a form of conventional medical practice that has been used in East Asia for thousands of years [18]. It stimulates specific acupoints to correct the imbalance of energy within the body. Owing to its nonpharmacological and minimally invasive advantages, acupuncture is commonly applied to various GI diseases including irritable bowel syndrome [19, 20], gastroparesis [21, 22], and constipation [23]. Several clinical trials have shown the potential effectiveness of acupuncture on GI function recovery, such as the bidirectional regulation effect on gastric myoelectrical activity and reduction of abdominal distension [24, 25].

Some previous meta-analyses evaluating the effectiveness of acupuncture in POI have been reported. A previous study [26], which included abdominal surgery patients, found that electroacupuncture (EA) or transcutaneous electric acupoint stimulation (TEAS) is effective for POI. Another two previous meta-analyses examined the effectiveness of acupuncture in cancer patients and showed that acupuncture and related therapies could improve the recovery of GI function [27, 28]. However, the evidence of acupuncture for POI is still inconclusive in GI

surgery patients. GI surgery is one of the most common types of major abdominal surgery and has a direct impact on the GI tract. Aiming to provide more targeted evidence for clinicians, we focused on POI patients undergoing GI surgery in this systematic review and meta-analysis to critically evaluate the effectiveness and safety of acupuncture.

## Materials and methods

The registered study protocol of this systematic review and meta-analysis is available in the PROSPERO International prospective register of systematic reviews database (https://www. crd.york.ac.uk/prospero/, identification number: CRD42020183593). We performed this study according to the Cochrane Handbook for Systematic Reviews of Interventions [29] and followed the Preferred Reporting Items for Systematic Reviews and Meta-Analyses Statement (PRISMA) guidelines [30].

### Database and search strategy

The following databases were searched up to May 30, 2021: PubMed, Embase, the Cochrane Library, China National Knowledge Infrastructure (CNKI), Wan fang Data, Chongqing VIP Database (CQVIP), and Chinese Biomedical Literature Database (CBM). The ClinicalTrials. gov was also searched to avoid omitting ongoing or unpublished studies. The reference lists of other systematic reviews and all included studies were used to obtain relevant studies. The following keywords were searched: "acupuncture", "acupuncture therapy", "electroacupuncture", "transcutaneous electrical acupoint stimulation", "postoperative ileus", "postoperative bowel disfunction", and "gastrointestinal surgery". The detailed search strategy is available in S1 Appendix. The literature regions and publication types were not restricted.

### Study selection

**Inclusion criteria.** (I) Participants: participants should be $> 18$ years old following GI surgery. (II) Interventions: acupuncture therapies, including manual acupuncture (MA), EA, TEAS, abdominal acupuncture and so on; (III) Comparisons: usual care or sham acupuncture. (IV) Outcomes: primary outcomes were time to first flatus (TFF) and time to first defecation (TFD), and secondary outcomes were time to bowel sounds recovery (TBSR) and length of hospital stay (LOS). (V) Study design: randomized controlled trial (RCT).

**Exclusion criteria.** (I) Non-GI surgery patients. (II) The acupuncture regimen was acupuncture combined with oral herbal medicine, embedding, acupoint injection, or other non-acupuncture-related therapy. (III) The comparison represented other techniques of traditional Chinese medicine, herbal medications, or other acupuncture styles. (IV) The primary outcome was insufficient.

### Data extraction

The data extraction process was independently performed by two reviewers (ZY and ZQ X). The two reviewers independently selected articles following the inclusion and exclusion criteria and assessed the full texts of the selected trials. The relevant information was extracted as follows: first author, year of publication, country, baseline characteristics of patients, number of patients, surgical procedures, intervention details, control types and main outcomes. All information was included in a standardized data extraction form. Divergence would be conquered by the adjudication of the supervisor (KW).

### Risk of bias assessment

The methodological qualities were assessed by two investigators (LK and YL) according to the Cochrane risk of bias (ROB) tool [29]. ROB was classified into three grades: low risk, high risk, or unclear risk. It included seven domains: (I) random sequence generation; (II) allocation concealment; (III) blinding of participants and personnel; (IV) blinding of outcome assessment; (V) selective reporting; (VI) incomplete outcome data; (VII) other bias. Disagreements were discussed between the two reviewers, and if these were unresolved, a third reviewer (JL) was added to the discussion until a consensus was reached.

### Evidence quality assessment

We used the Grading of Recommendations Assessment, Development, and Evaluation (GRADE) approach to rate the overall quality of evidence [31]. The GRADE guideline has five domains, including the risk of bias, inconsistency, indirectness, imprecision, and potential publication bias. GRADE provides four levels of quality (high, moderate, low, and very low) for evidence grading. Two researchers (ZY and XQ W) performed the assessment process independently, and a third researcher (KW) then reviewed the evalution. Any disagreement was resolved by discussion with professional specialist advice.

### Data analysis

We used Review Manager (RevMan) version 5.4.1 software (The Nordic Cochrane Center, The Cochrane Collaboration, Copenhagen, Denmark [32]) to perform the meta-analysis. The continuous variables were assessed by mean difference (MD) with a 95% confidence interval (CI) when the unit was the same. Otherwise, the standardized mean difference (SMD) was used. A P value of $< 0.05$ was considered statistically significant. The magnitude of the effect size of SMD was rated as follows: $\leq 0.2$ indicated a small effect, 0.5 indicated a moderate effect, and $\geq 0.8$ indicated a large effect. As for the testing of heterogeneity, we used $\chi^2$ test. If $I^2$ was $< 50\%$ in the results, we selected a fixed-effects model to pool the data. Otherwise, a random-effects model was adopted.

To explore the source of heterogeneity, subgroup analysis was conducted to analyze the primary outcomes. Sensitivity analysis was performed by excluding each RCT sequentially and comparing the model characteristics to test the robustness of the result. A funnel plot was used to detect publication bias when at least 10 trials were included. Additionally, we used Stata 12.0 software (StataCorp, Texas, USA) to put Egger's test aiming to assess the funnel plot. The significant publication bias was defined as a P-value of $< 0.1$.

## Results

### Search results

We initially identified 430 studies through database searching and eight studies through other sources. After removing 102 duplicates, 336 articles were screened, and 251 articles were removed by titles and abstract. Furthermore, 67 studies were excluded after reviewing full texts based on eligibility criteria. Finally, 18 studies [33–50] were included in this study (Fig 1).

### Study characteristics

Included RCTs were published from 2008 to 2020. Sixteen trials [33–39, 41–48, 50] were from China, one trial [40] was from Korea, and one trial [49] was from America. The study size ranged from 30 to 165. The main characteristics of the studies are summarized in Table 1.

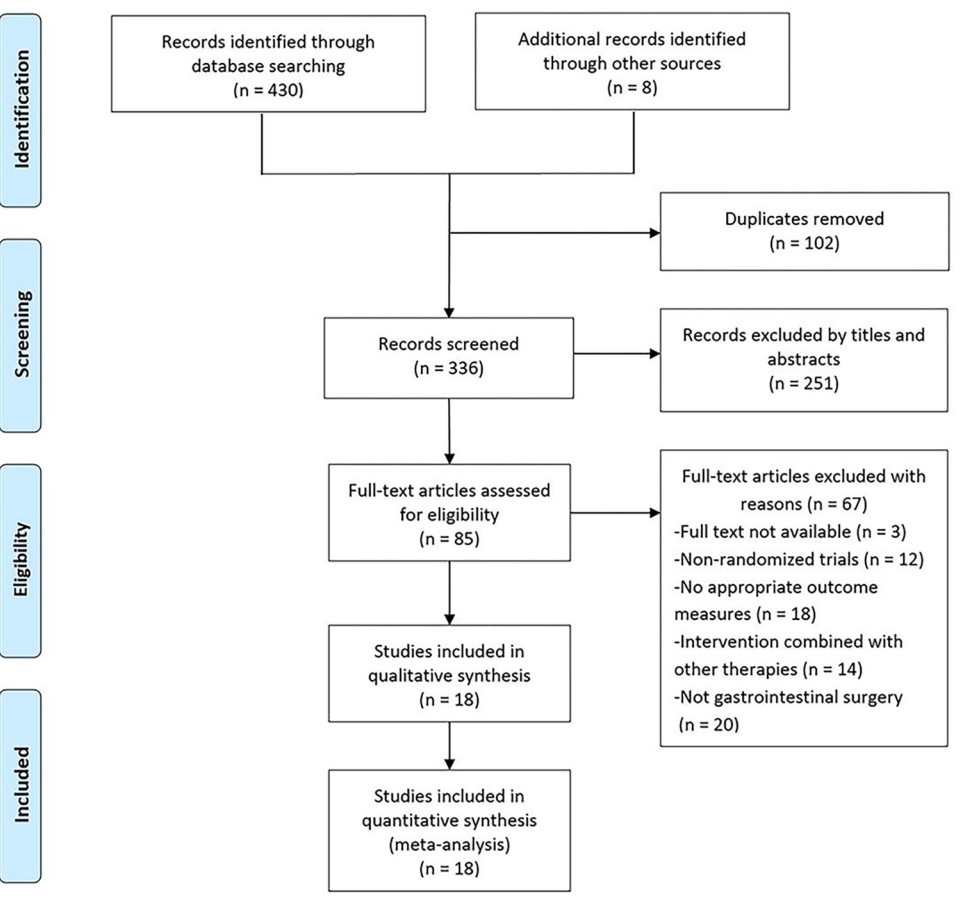

**Fig 1. The flow diagram of study selection.**

Eighteen studies included 1413 participants, of whom 666 were in the intervention group, and 747 were in the control group. There were 845 males and 568 females among these participants. The age of participants ranged from 28.55 years to 68.5 years.

In terms of intervention, three types of acupuncture techniques were involved: MA, EA, and TEAS. Among all studies, a total of 23 acupoints were used. The commonly used acupoints were ST36, ST37, and SP6, which were located in the stomach meridian and spleen meridian. All trials [33–50] reported the treatment timing. Two trials [35, 37] performed acupuncture before and after the operation, respectively. Other trials [33, 34, 36, 38–50] all performed acupuncture after the operation. Eleven studies [33–35, 38, 41–43, 45, 47–49] reported *deqi* sensation. The acupuncture retention time varied from 20 to 60 minutes. The frequency of treatment comprised once per day (1/d), twice per day (2/d), or three times per day (3/d). The main details of the interventions are summarized in Table 2.

In terms of control, 14 studies [33, 34, 36, 38–40, 42–44, 46–50] used usual care, and three studies [35, 37, 41] applied sham acupuncture plus usual care. Additionally, there was one study [45] that set up two control groups including usual care and sham acupuncture.

The main outcomes of most trials were TFF, TFD, TBSR, and LOS. Additionally, there were trials reporting the analgesic consumption, postoperative nausea and vomiting, and time to nasogastric tube removal.

**Table 1. Characteristics of included studies in the systematic review and meta-analysis.**

| Study | Country | Sample size (M/F) | Ages (I/C) | Surgical approach | Control a | Intervention | Course of treatment | Outcomes |
|---|---|---|---|---|---|---|---|---|
| Xu 2020 [33] | China | I: 16/14 C: 15/15 | I: 28.55 ± 9.75 C: 29.64 ± 9.34 | Open | UC | EA | NA | TFF, TBSR, LOS |
| Gu 2019 [35] | China | I: 31/27 C: 29/30 | I: 57.59 ± 7.32 C: 56.67 ± 6.23 | Laparoscopic | SA | TEAS | From 30 min before anesthetic induction to 2 days after operation | TFF, TFD, TBSR, VAS, AC, Incidence of PONV, Patient satisfaction |
| Pu 2019 [34] | China | I: 20/11 C: 16/15 | I: 55.07 ± 10.17 C: 56.13 ± 11.08 | Open | UC | EA | 5 consecutive postoperative days | TFF, TFD, TBSR, LOS, PSA |
| Chen 2018 [36] | China | I: 22/11 C: 24/6 | I: 63.0 ± 9.70 C: 59.0 ± 8.30 | Open, laparoscopic | UC | TEAS | Starting on postoperative day 1, for 5 consecutive days or until passing flatus | TFF, TFD, LOS, TNTR, TLSD, PSA |
| Kang 2017 [39] | China | I: 39/29 C: 41/27 | I: 41.28 ± 10.36 C: 40.75 ± 10.19 | Unclear | UC | EA | NA | TFF, TFD, TBSR, LOS, ADL |
| Jung 2017 [40] | Korea | I: 16/2 C: 16/2 | I: 60.94 ± 9.43 C: 60.06 ± 13.18 | Open, laparoscopic | UC | EA | 5 consecutive postoperative days | TFF, TFD, LOS, SWI, SSD, the number of remnant sitz markers in the small intestine on abdominal radiograph |
| Yuan 2017 [37] | China | I: 12/18 C: 16/14 | I: 53.9 ± 9.8 C: 54.6 ± 10.4 | Laparoscopic | SA | TEAS | Half an hour before operation, then 3 consecutive postoperative days | TFF, TBSR, LOS, Incidence of PONV, AC |
| Qian 2017 [38] | China | I: 20/10 C: 17/13 | I: 59 ±10 C: 60 ± 11 | Unclear | UC | MA | 7 consecutive postoperative days | TBSR, TFF, LOS, TGTR, IPAR, QOL, Hospitalization expenses |
| Zhang 2014 [41] | China | I: 11/8 C: 11/9 | I: 63 ± 9 C: 60 ± 10 | Open | SA | EA | 30 min after operation, then 4 consecutive postoperative days | TFF, TFD, TBSR, |
| Xiao 2014 [42] | China | I: 18/12 C1: 19/11 C2: 14/16 | I: 55.87 ± 10.49 C1: 55.33 ± 10.83 C2: 54.63 ± 10.25 | Open | UC | I1: EA I2: MA | 5 consecutive postoperative days | TFF, TFD, TBSR, PSA |
| Tong 2014 [43] | China | I: 24/18 C: 26/16 | I: 58.6 ± 15.1 C: 59.2 ± 14.7 | Open | UC | MA | 2 consecutive postoperative days | TFF, TFD, TBSR, TLFI, TNTR, PSA |
| Wang 2013 [44] | China | I: 41/23 C: 43/21 | I: 40.5 ± 9.7 C: 42.3 ± 9.1 | Unclear | UC | MA | 10 consecutive postoperative days | TFF, TFD, TBSR |
| Ng 2013 [45] | China | I: 35/20 C1: 33/22 C2: 31/24 | I: 67.4 ± 9.7 C1: 67.4 ± 10.7 C2: 68.5 ± 10.6 | Laparoscopic | C1: UC C2: SA | EA | Starting on postoperative day 1, for 4 consecutive days or until the first defecation | TFF, TFD, LOS, TTSD, TWI, VAS, AC |
| Shi 2012 [46] | China | I: 15/15 C: 17/13 | I: 53.17 ± 13.491 C: 53.77 ± 13.320 | Unclear | UC | EA | 3 consecutive postoperative days | TFF, TFD, TBSR, PSA |
| Yang 2011 [47] | China | I: 21/10 C: 18/11 | I: 60.9 ± 6.63 C: 62 ± 6.98 | Unclear | UC | EA | Starting on postoperative day 1 until 3 days after first defecation | TFF, TFD, TBSR |
| Wang 2011 [48] | China | I: 11/4 C: 9/6 | I: 58.0 ± 10.24 C: 60.4 ± 11.01 | Open | UC | MA | 5 consecutive postoperative days | TFF, TFD, TBSR, PSA |
| Meng 2010 [50] | China | M: 47 F: 38 | I: 54.3 (mean) C: 53.1(mean) | Unclear | UC | EA | Starting on postoperative day 1, for 6 consecutive days or until the first bowel movement | TFF, TFD, EGEG, QOL |

(*Continued*)

**Table 1.** (Continued)

| Study | Country | Sample size (M/F) | Ages (I/C) | Surgical approach | Control [a] | Intervention | Course of treatment | Outcomes |
|---|---|---|---|---|---|---|---|---|
| Garcia 2008 [49] | America | I: 25/13 C: 26/14 | 35–45: 14 46–55: 24 56–65: 26 66–75: 12 75: 1 | Unclear | UC | EA | Starting on postoperative day 1, for 4 consecutive days or until the first defecation | TFF, TFD, LOS, AC, QOL |

[a] Usual care included: routine nasogastric tubes, intravenous fluids, parenteral nutrition and early mobilization.

Abbreviations: I/C, Intervention group/Control group; M/F, Male/Female; MA, manual acupuncture; EA, electroacupuncture; TEAS, transcutaneous electrical acupoint stimulation; UC, usual care; SA, sham acupuncture; ERAS, enhanced recovery after surgery; NA, not available; TFF, time to first flatus; TBSR, time to bowel sounds recovery; LOS, length of hospital stay; TFD, time to first defecation; VAS, visual analogue scale; AC, analgesic consumption; PONV, postoperative nausea and vomiting; PSA, postoperative symptom assessment; TNTR, time to nasogastric tube removal; TLSD, time to liquid and semi-liquid diet; ADL, activities of daily living; SWI, start of water intake; SSD, start of soft diet; TGTR, time to gastric tube removal; IPAR, incidence of postoperative adverse reactions; QOL, quality of life status, including pain, nausea, insomnia, abdominal distention and general sense of well-being; TLFI, time to liquid food intake; TTSD, time to tolerated a solid diet; TWI, time to walk independently; EGEG, electro-gastroenterography.

### Risk of bias assessment

The risk of bias of each included trial is listed in Fig 2. All studies were described as "randomized" but five [39, 40, 43, 44, 46] did not describe details of sequence generation; hence, these were judged as "unclear risk". Five studies [34, 35, 37, 41, 45] described the method of allocation concealment; therefore, these were judged as "low risk". Thirteen trials [33, 36, 38–40, 42–44, 46–50] did not offer the details of the allocation concealment; hence, we judged them as "unclear risk". It was not possible for practitioners to be blinded in the acupuncture treatment. Therefore, we judged two trials [35, 45] as "low risk" of performance bias since the outcome investigator was blind to group allocation, meaning the acupuncture performer did not participate in the data collection. Thirteen trials [36–44, 46–49] did not use the blind method; thus, we judged these trials as "high risk". Furthermore, three trials [33, 34, 50] did not mention the details of the blinding method; therefore, we judged them as "unclear risk". Three trials [35, 37, 45] described the blinding of the outcome assessor; hence, these were judged as "low risk". Fifteen trials [33, 34, 36, 38–44, 46–50] did not adequately describe whether the outcome assessor was blinded to the treatment allocation; therefore, these were judged as "unclear risk". Three trials [35, 40, 50] had a high risk of attrition bias due to participants' withdrawal from the studies. All the trials reported the predetermined outcome measures; hence, the reporting bias were judged as "low risk". Four trials [33, 39, 43, 44] were judged at unclear risk of other potential bias due to insufficient registration information.

### Overall effectiveness of acupuncture

**Acupuncture versus usual care.** *Time to first flatus*. Fifteen trials [33, 34, 36, 38–40, 42–50] involving 1162 participants evaluated the change in TFF (hour). Pooled results indicated that acupuncture had a better effect in shortening the TFF compared to usual care (SMD = -1.14; 95% CI: -1.54 to -0.73; P < 0.00001; I$^2$ = 90%; Fig 3A).

*Time to first defecation*. Twelve trials [34, 39, 40, 42–50] with 980 participants examined the change in TFD (hour). The analysis data showed that acupuncture resulted in a reduction in TFD compared to usual care (SMD = -1.31; 95% CI: -1.88 to -0.74; P < 0.00001; I$^2$ = 94%; Fig 3B).

*Time to bowel sounds recovery*. Ten trials [33, 34, 38, 39, 42–44, 46–48] involving 800 participants reported this outcome and showed significant shortening of TBSR (hour) in the

**Table 2. Details of intervention.**

| Trials | Intervention | Acupoints selection | Starting intervention | Frequency | Response sought | Retention time | Stimulus parameter |
|---|---|---|---|---|---|---|---|
| Xu 2020 [33] | EA | ST36, SP6, PC6, CV12 | Postoperative 6h | 2/d | DS | 20 min | Dilatational wave, 2Hz |
| Gu 2019 [35] | TEAS | ST36 and PC6 | 1.30 min before anesthetic induction 2. Postoperative day 1 | 3/d | DS | 1.60 min 2.30 min | 5−30 mA |
| Pu 2019 [34] | EA | PC6, SP4, ST37, ST36 | Postoperative day 1 | 1/d | DS | 30 min | Discontinuous wave, 2 Hz, 1−10 mA |
| Chen 2018 [36] | TEAS | ST36 and PC6 | Postoperative day 1 | 2/d | NA | 60 min | ST36: 2 s on, 3 s off, 25 Hz, 0.5 ms, 2−6 mA; PC6: 0.1 s on, 0.4 s off, 100 Hz, 0.25ms, 2−6 mA. |
| Kang 2017 [39] | EA | ST36 | Postoperative day 1 | 1/d | NA | 30 min | Continuous wave |
| Jung 2017 [40] | EA | ST36, SP6, LI4, SJ6, LV3, LI11 GV20†, EX-HN3†, GV26†, CV24† | Postoperative day 1 | 1/d | NA | 25−30 min | ST36, SP6, LI4, TE6:100 Hz |
| Yuan 2017 [37] | TEAS | PC6, LI4, ST36 | 1. Preoperative 30 min 2. Postoperation | 1/d | NA | 30min | NA |
| Qian 2017 [38] | MA | ST36, ST37, ST39 | Postoperative day 1 | 1/d | DS | 20 min | NA |
| Zhang 2014 [41] | EA | ST36 | Postoperative 30 min | 1/d | DS | 30 min | 2 Hz, 0.16 ms |
| Xiao 2014 [42] | EA | ST36, ST37 | Postoperative day 1 | 1/d | DS | 20 min | Dilatational wave |
| Tong 2014 [43] | MA | ST36, SP6, ST37, SP4 | Postoperative 2 h | 1/d | DS | 30min | NA |
| Wang 2013 [44] | MA | CV12†, CV10†, CV6†, CV4†; ST25, SP15, ST26, Xiafengshidian | Postoperation | 1/d | NA | 30 min | NA |
| Ng 2013 [45] | EA | ST36, SP6, LI4, SJ6 | Postoperative day 1 | 1/d | DS | 20 min | 100 Hz |
| Shi 2012 [46] | EA | ST36, ST37, ST39 | Postoperative 6 h | 2/d | NA | 30 min | Dilatational wave 4 Hz/20 Hz |
| Yang 2011 [47] | EA | ST36, ST37, ST39 | Postoperative day 1 | 1/d | DS | 30 min | Continuous wave |
| Wang 2011 [48] | MA | PC6, SP4, ST37, ST36 | Postoperative 24h | 1/d | DS | 30 min | NA |
| Meng 2010 [50] | EA | SJ6, GB34, ST36, ST37 | Postoperative day 1 | 1/d | NA | 20 min | SJ6, GB34: Continuous wave, 2 Hz |
| Garcia 2008 [49] | EA | LI4, SP6, ST36, ST25, CV6†, CV12† | Postoperative day 1 | 2/d | DS | 20 min | LI 4 (positive) to ST 36 (negative), 50 Hz, 16 mA |

†unilaterally

Abbreviations: MA, manual acupuncture; EA, electroacupuncture; TEAS, transcutaneous electroacupuncture stimulation; ST36, zusanli acupoint; SP6, sanyinjiao acupoint; PC6, neiguan acupoint; CV12, zhongwan acupoint; SP4, gongsun acupoint; ST37, shangjuxu acupoint; LI4, hegu acupoint; SJ6, zhigou acupoint; LV3, taichong acupoint; LI11, quchi acupoint; GV20, baihui acupoint; EX-HN3, yintang acupoint; GV26, shuigou acupoint; CV24, chengjiang acupoint; ST39, xiajuxu acupoint; CV10, xiawan acupoint; CV6, qihai acupoint; CV4, guanyuan acupoint; ST25, tianshu acupoint; SP15, daheng acupoint; ST26, wailing acupoint; GB34, yanglingquan acupoint; DS, de qi sensation, the achievement of a radiating sensation with paresthesia was indicative of effective needling; NA, not available.

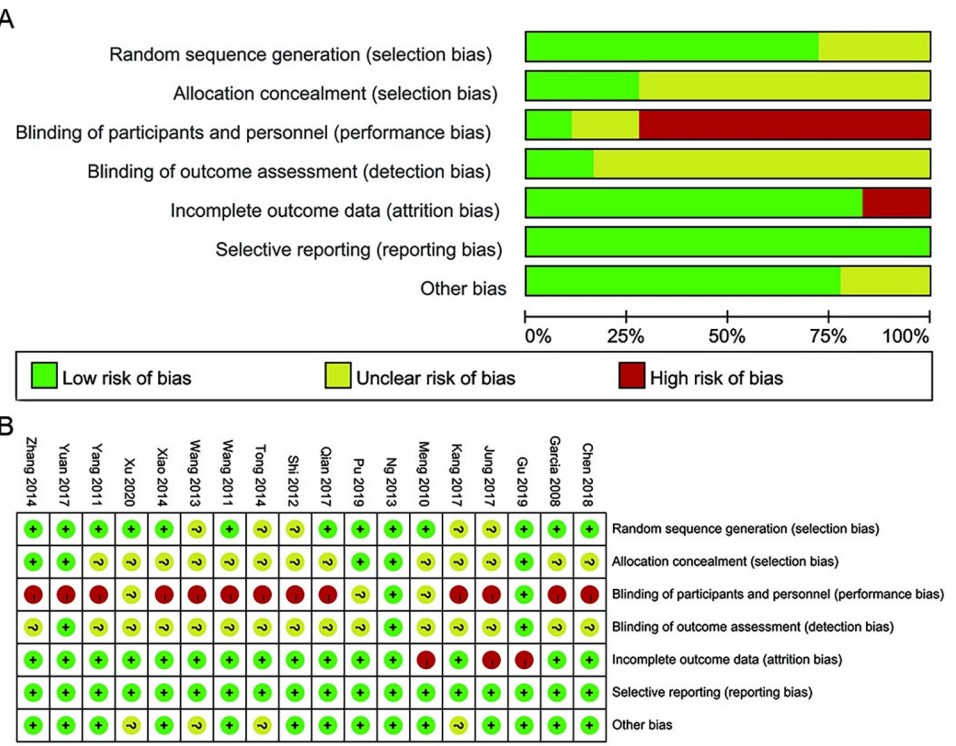

**Fig 2. Risk of bias summary.**

acupuncture group compared to the usual care group (SMD = -1.57; 95% CI: -2.14 to -1.01; P < 0.00001; $I^2$ = 91%; Fig 3C).

*Length of hospital stay.* Eight trials [33, 34, 36, 38–40, 45, 49] involving 605 participants reported the LOS (day). The LOS was shorter in the acupuncture group compared to the usual care group (MD = -1.68d; 95% CI: -2.55 to -0.80; P = 0.0002; $I^2$ = 86%; Fig 3D).

**Acupuncture versus sham acupuncture.** *Time to first flatus.* Four trials [35, 37, 41, 45] involving 326 participants evaluated the change in TFF. The results showed a difference between the acupuncture group and the sham acupuncture group (SMD = -0.81; 95% CI: -1.40 to -0.23; P = 0.007; $I^2$ = 83%; Fig 4A).

*Time to first defecation.* Three trials [35, 41, 45] with 266 participants evaluated the change in TFD. The analysis data showed that acupuncture had a better effect in reducing TFD compared to sham acupuncture (SMD = -0.34; 95% CI: -0.58 to -0.10; P = 0.006; $I^2$ = 0%; Fig 4B).

*Time to bowel sounds recovery.* Three trials [35, 37, 41] involving 216 participants reported TBSR and showed more reduction in the acupuncture group compared to the sham acupuncture group (SMD = -1.03; 95% CI: -1.64 to -0.43; P = 0.0008; $I^2$ = 74%; Fig 4C).

*Length of hospital stay.* Three trials [37, 41, 45] involving 209 participants reported the LOS. The result showed that there was no statistical difference between the acupuncture group and the sham acupuncture group (MD = -0.99 d; 95% CI: -2.06 to -0.08; P = 0.07; $I^2$ = 64%; Fig 4D).

## Subgroup analysis

Due to the limited number of studies, we only analyzed the primary outcomes TFF and TFD in the comparison between acupuncture and usual care. The subgroups were based on the

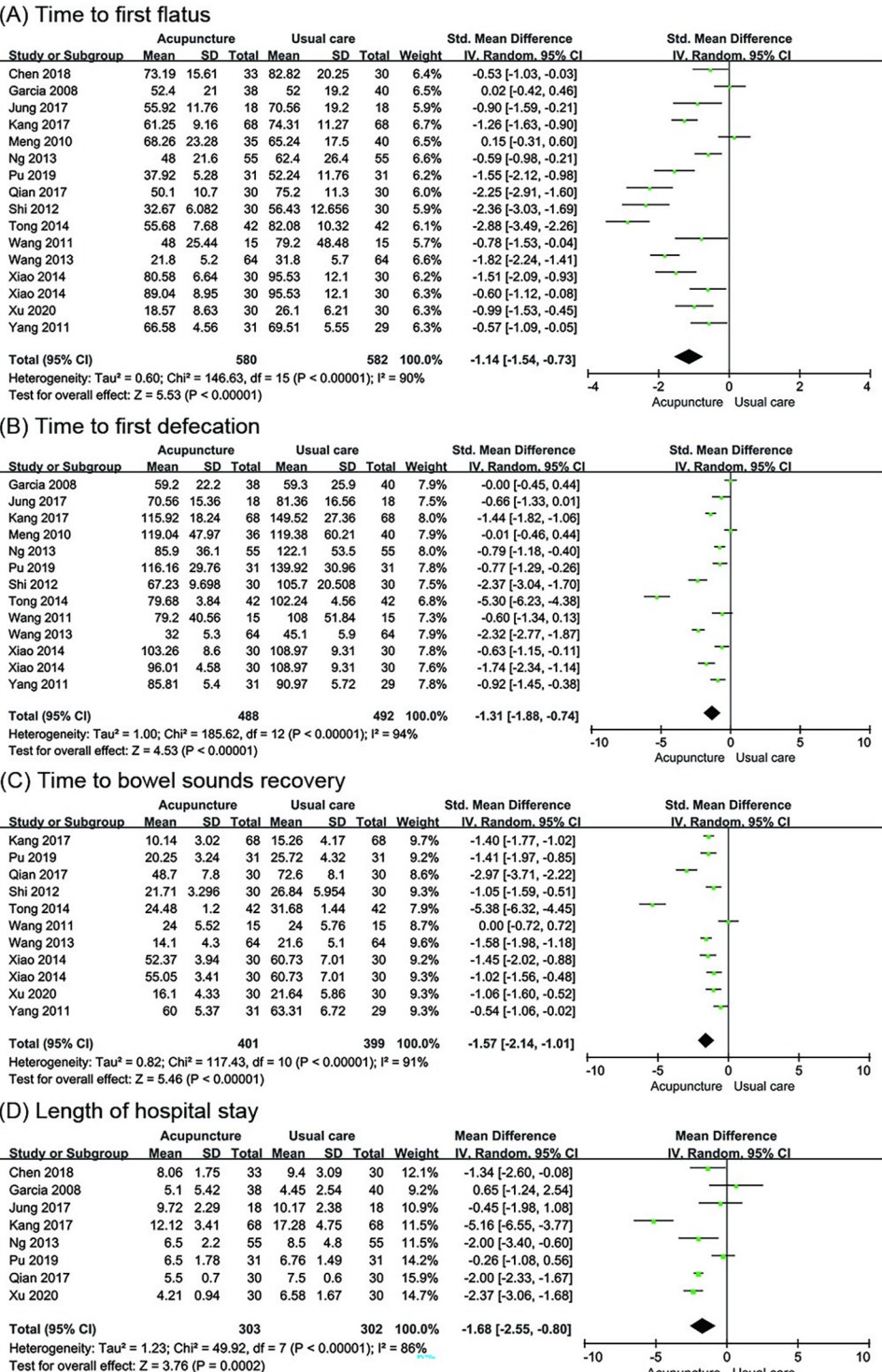

**Fig 3.** Meta-analysis of acupuncture versus usual care for (A) Time to first flatus, (B) Time to first defecation, (C) Time to bowel sounds recovery and (D) Length of hospital stay.

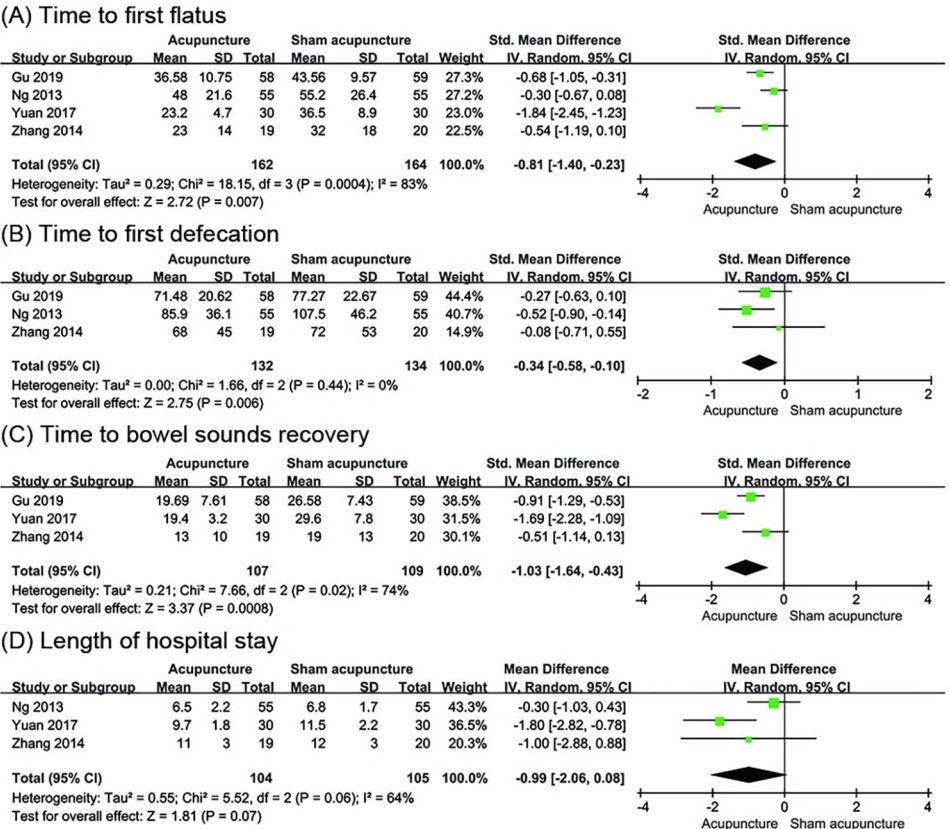

**Fig 4.** Meta-analysis of acupuncture versus sham acupuncture for (A) Time to first flatus, (B) Time to first defecation, (C) Time to bowel sounds recovery and (D) Length of hospital stay.

following characteristics: (I) acupuncture technique: MA, EA, or TEAS; (II) acupoints combination: distal acupoints combination or distal–proximal acupoints combination; (III) frequency of treatment sessions: 1 session per day (1/d) or 2 sessions per day (2/d). The results are listed in the Table 3.

The subgroup analysis showed that studies with all types of acupuncture techniques had a significant effect on reducing TFF and TFD substantially. Regarding the type of acupoints combination, studies that applied distal acupoints combination showed significant improvement in reducing TFF (SMD = -1.19; 95% CI: -1.64 to -0.74; P < 0.00001; $I^2$ = 89%) and TFD (SMD = -1.32; 95% CI: -1.92 to -0.73; P < 0.0001; $I^2$ = 93%). In the analysis based on the frequency of treatment sessions, both two frequencies showed a significant effect in reducing TFF. However, only acupuncture treatment with 1 session per day showed significant improvement in reducing TFD (SMD = -1.32; 95% CI: -1.93 to -0.72; P < 0.0001; $I^2$ = 93%). No factors could account for the heterogeneity.

## Sensitivity analysis

In comparison with usual care, there were no changes in the significant outputs from the meta-analysis by omitting a single study. These heterogeneities did not influence the stability of the result. In comparison with sham acupuncture, there were changes in the outputs after excluding each study. After removing the study conducted by Gu et al. [35], the results of TFF (P = 0.06), TFD (P = 0.07), and TBSR (P = 0.06) showed no significance. After excluding the

**Table 3. Subgroup analysis.**

| Outcome | Subgroup | Studies | Patients | Effect Sizes SMD | 95%CI | Heterogeneity I² (%) | P value |
|---|---|---|---|---|---|---|---|
| Time to first flatus | **Acupuncture technique** | | | | | | |
| | MA | 5 | 362 | -1.67 | [-2.47, -0.86] | 90 | < 0.0001 |
| | EA | 10 | 737 | -0.94 | [-1.38, -0.49] | 87 | < 0.0001 |
| | TEAS | 1 | 63 | -0.53 | [-1.03, -0.03] | - | = 0.04 |
| | **Acupoints combination** | | | | | | |
| | Distal acupoints combination | 12 | 896 | -1.19 | [-1.64, -0.74] | 89 | < 0.00001 |
| | Distal–proximal acupoints combination | 3 | 266 | -0.93 | [-2.05, 0.19] | 94 | = 0.1 |
| | **Frequency of treatment session** | | | | | | |
| | 1/d | 11 | 901 | -1.2 | [-1.66, -0.75] | 89 | < 0.00001 |
| | 2/d | 4 | 261 | -0.94 | [-1.85, -0.03] | 92 | = 0.04 |
| Time to first defecation | **Acupuncture technique** | | | | | | |
| | MA | 4 | 302 | -2.18 | [-3.87, -0.50] | 97 | = 0.01 |
| | EA | 9 | 678 | -0.93 | [-1.40, -0.46] | 88 | = 0.0001 |
| | **Acupoints combination** | | | | | | |
| | Distal acupoints combination | 10 | 774 | -1.32 | [-1.92, -0.73] | 93 | <0.0001 |
| | Distal–proximal acupoints combination | 2 | 206 | -1.16 | [-3.43, 1.11] | 98 | = 0.32 |
| | **Frequency of treatment session** | | | | | | |
| | 1/d | 10 | 842 | -1.32 | [-1.93, -0.72] | 93 | <0.0001 |
| | 2/d | 2 | 138 | -1.17 | [-3.49, 1.14] | 97 | = 0.32 |

Abbreviations: MA, manual acupuncture; EA, electroacupuncture; TEAS, transcutaneous electrical acupoint stimulation; SMD, standardized mean difference; 95%CI, 95% Confidence interval.

study conducted by Ng et al. [45], the result of TFD (P = 0.17) showed no significance, and the result direction of LOS (P = 0.0004) was reversed. After removing the study of Yuan et al. [37], the heterogeneity was significantly reduced, and the result was not altered (see S2 Appendix).

## Adverse events

There were four studies [34, 36, 49, 50] that reported the information on adverse events. Only one study [36] reported mild bruising of the wrist due to TEAS. Three studies [34, 49, 50] stated that there were no adverse events for acupuncture.

## Publication bias

The funnel plot of 15 trials included in the meta-analysis for TFF (Fig 5A) showed that these were approximately symmetric. Additionally, the funnel plot of 12 trials reported for TFD (Fig 5B) showed a similar tendency. Egger's test demonstrated that there was no obvious publication bias (TFF: P = 0.171; TFD: P = 0.14) (Fig 6).

## Level of overall evidence

Table 4 displays a summary of the overall certainty in evidence for the effectiveness of acupuncture on the relevant outcomes. In the comparison of acupuncture with usual care, the evidence indicated with a low level of certainty that acupuncture was associated with reducing TFF, TFD, and TBSR. A moderate level of certainty suggested that acupuncture was associated with reducing LOS as compared with usual care. Certainty in the evidence for the comparison of acupuncture with sham acupuncture was variable in TFF (very low), TFD (low), TBSR (very low), and LOS (low).

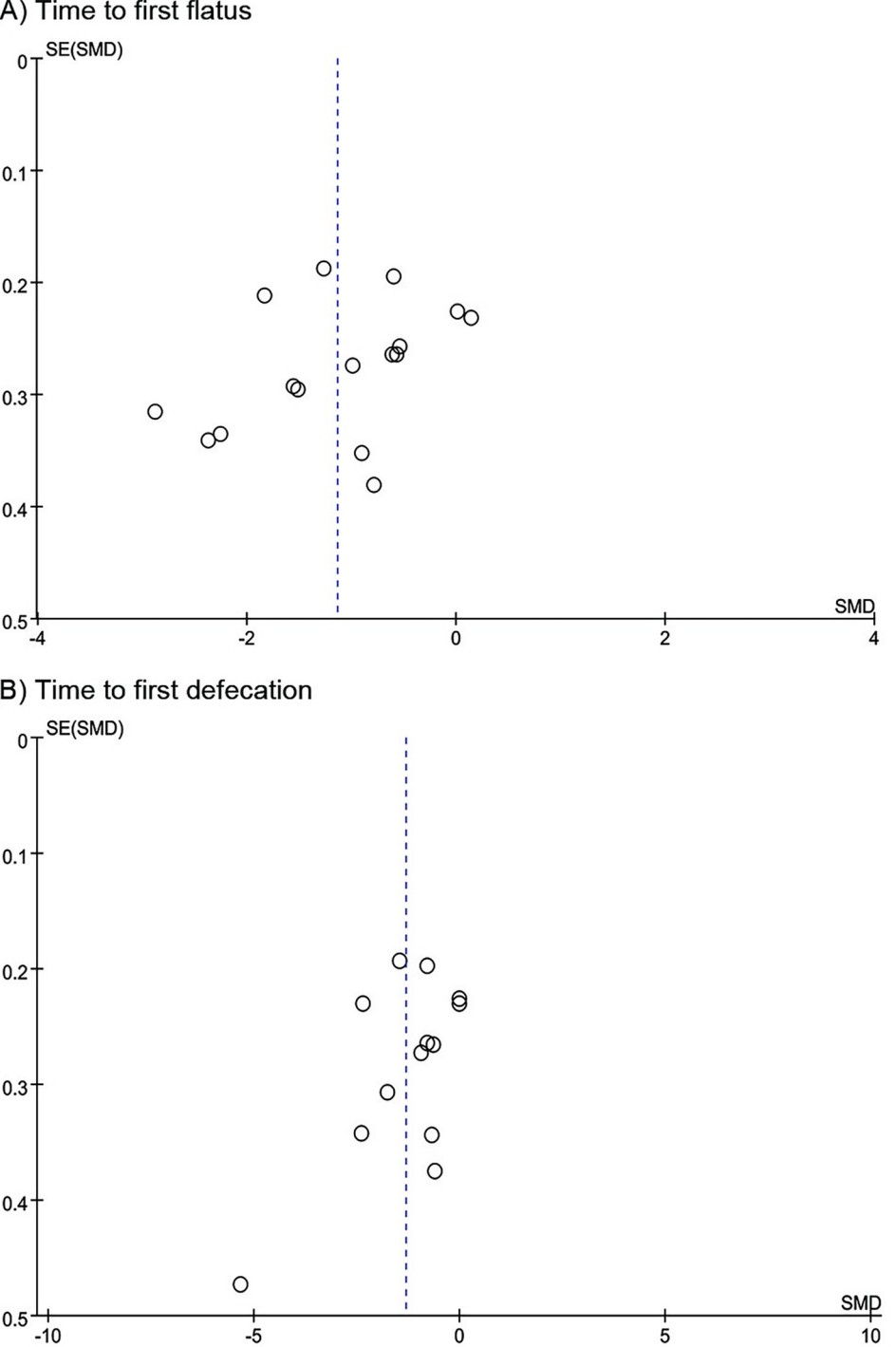

**Fig 5.** Funnel plot of acupuncture versus usual care for (A) Time to first flatus and (B) Time to first defecation.

## Discussion

### Main findings

This systematic review and meta-analysis aimed to assess the effectiveness and safety of acupuncture for POI among patients undergoing GI surgery. Low to moderate quality of evidence

## (A) Time to first flatus

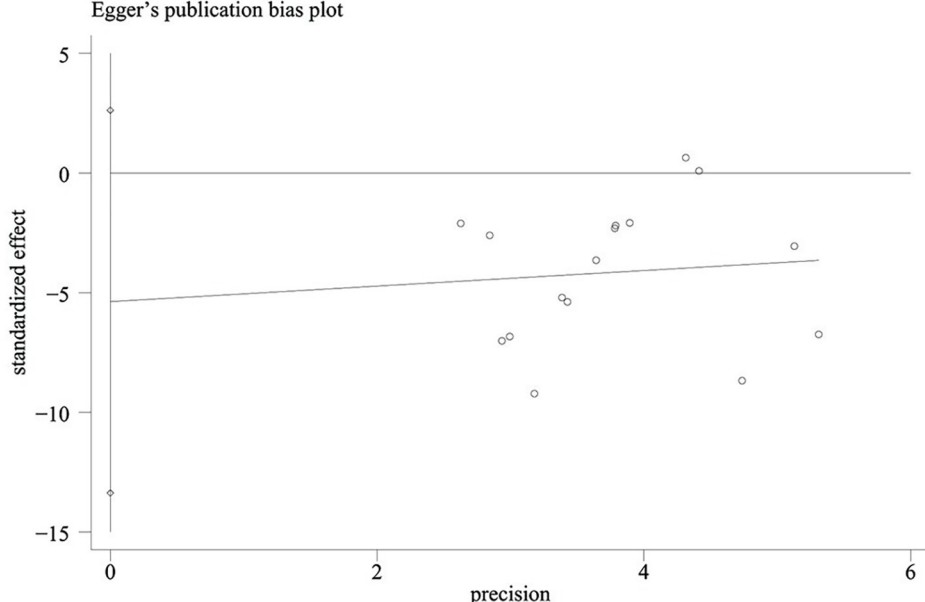

## (B) Time to first defecation

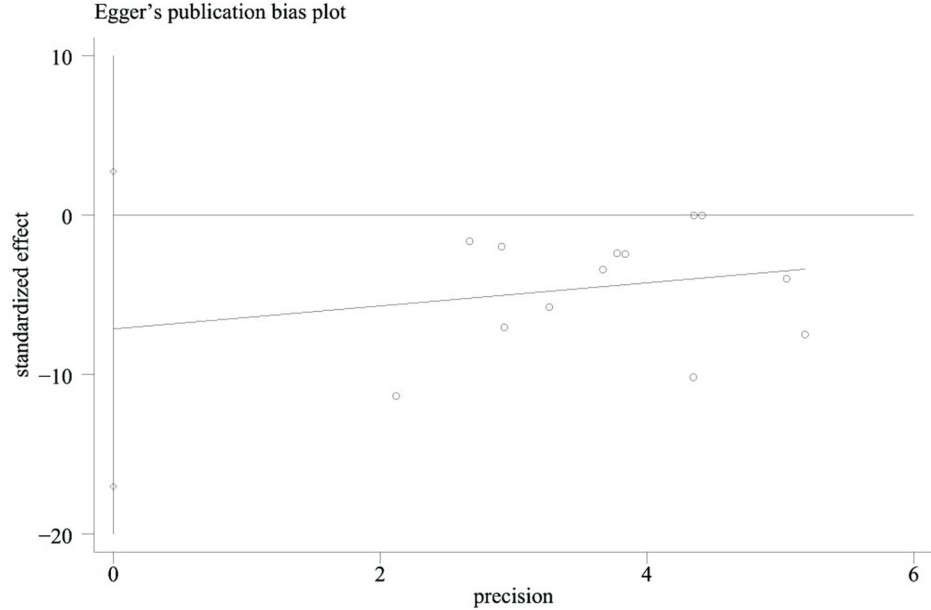

**Fig 6.** Egger's test of acupuncture versus usual care for (A) Time to first flatus and (B) Time to first defecation.

showed that acupuncture could reduce the TFF, TFD, TBSR, and LOS compared with usual care. The subgroup analysis indicated that acupuncture treatment with distal acupoints combination and frequency of 1 session per day had effectiveness in reducing TFF and TFD. The sensitivity analysis and the publication bias supported the stability of the overall effect size. Very low to low-quality evidence suggested that acupuncture had an effect on reducing TFF, TFD, and TBSR compared with sham acupuncture. However, this result should be interpreted

**Table 4. The overall evidence quality for outcome measure.**

| Group | N | Participants | | Absolute effect [95%CI] | Certainty assessment | | | | | Certainty | Importance |
|---|---|---|---|---|---|---|---|---|---|---|---|
| | | I | C | | Risk of bias | Inconsistency | Indirectness | Imprecision | Other considerations | | |
| Acupuncture vs. Usual care | Time to First Flatus | | | | | | | | | | |
| | 15 | 580 | 582 | SMD-1.14 [-1.54, -0.73] | Serious[a] | Serious[a] | No | No | No | Low | Critical |
| | Time to First Defecation | | | | | | | | | | |
| | 12 | 488 | 492 | SMD-1.31 [-1.88, -0.74] | Serious[a] | Serious[a] | No | No | No | Low | Critical |
| | Time to Bowel Sounds Recovery | | | | | | | | | | |
| | 10 | 401 | 399 | SMD-1.57 [-2.14, -1.01] | Serious[a] | Serious[a] | No | No | No | Low | Critical |
| | Length of hospital stay | | | | | | | | | | |
| | 8 | 303 | 302 | MD-1.68 [-2.55, -0.80] | No | Serious[a] | No | No | No | Moderate | Importance |
| Acupuncture vs. Sham acupuncture | Time to First Flatus | | | | | | | | | | |
| | 4 | 162 | 164 | SMD-0.81 [-1.40, -0.23] | Serious[a] | Serious[a] | No | Serious[c] | No | Very Low | Critical |
| | Time to First Defecation | | | | | | | | | | |
| | 3 | 132 | 134 | SMD-0.34 [-0.58, -0.10] | Serious[a] | No | No | Serious[c] | No | Low | Critical |
| | Time to Bowel Sounds Recovery | | | | | | | | | | |
| | 3 | 107 | 109 | SMD-1.03 [-1.14, -0.43] | Serious[a] | Serious[a] | No | Serious[c] | No | Very Low | Critical |
| | Length of hospital stay | | | | | | | | | | |
| | 3 | 104 | 105 | MD-0.99 [-0.26, 0.08] | No | Serious[a] | No | Serious[c] | No | Low | Importance |

Abbreviations: N, No. of studies; I, intervention; C, control; CI, confidence interval; MD, mean difference; SMD, standard mean difference.

a. Downgraded due to serious risk of bias: high risk of performance bias and unclear risk of selection bias and detection bias.

b. Downgraded due to substantial heterogeneity.

c. Downgraded due to small sample size.

with caution since the sensitivity analysis indicated that the result was not stable. Relatively few trials reported information on adverse events from acupuncture; hence, the overall safety should be further validated.

## Quality of the evidence

The overall quality of evidence for related outcomes was very low to moderate. In this meta-analysis, the serious risk of bias was the main problem in the included trials. It was mostly related to the deficient report of blinding and uncertainties about the allocation concealment, which have a potential impact on exaggerating the true effect size of acupuncture. Actually, it is challenging to have a low risk of bias in blinding in acupuncture clinical trials. First, it is infeasible to make the acupuncturist blinded to patients due to the nature of acupuncture intervention [51]. Second, participants who have experience with acupuncture are difficult to be blinded due to their general recognition of acupuncture. Therefore, future RCTs should pay more attention to how to make participants not being able to distinguish the real acupuncture from a sham control [52]. Moreover, the allocation concealment should be adequately reported.

## Acupoints combination

Based on the theory of traditional Chinese acupuncture, the acupoints combination is the key to ensuring the comprehensive curative effects of acupuncture. Distal–proximal acupoints combination and distal acupoints combination are the two basic methods for combining acupoints. Compared with previous studies, we assessed the effectiveness of acupuncture for POI using the subgroup analysis of the different acupoints combination for the first time in this review. The distal acupoints refer to acupoints distant from the abdomen of the GI district, and proximal acupoints are defined as acupoints on the abdomen closer to the GI district. There were three studies [33, 44, 49] that applied distal–proximal acupoints combination and

fifteen studies [34–43, 45–48, 50] that applied distal acupoints combination. In both two types of acupoints combination, ST36 was the main acupoint which located below the knee and on the tibialis anterior muscle. ST36 showed tropism for all GI disorders and had great efficacy in clinical practice, making it one of the master acupoints in Traditional Chinese Medicine [53]. Additionally, most studies on the distal acupoints combination chose lower limb acupoints. There was also one study [40] that combined scalp acupoints with limb acupoints. In the distal–proximal acupoints combination, the proximal acupoints included CV12, CV10, CV6, and CV4, which all were located on the midline of the abdomen.

Surprisingly, we found that acupuncture treatment with distal acupoints combination showed effectiveness in reducing TFF and TFD compared with usual care. However, the distal–proximal acupoints combination showed no statistical difference. Given the potential injuries to the abdomen from GI surgery, one possible reason is that applying acupuncture on proximal acupoints located on the abdomen may cause discomfort to the patient. Furthermore, it may have the risk of infection on the incision site when applying acupuncture on proximal acupoints. Due to the limited number of studies, the effectiveness and safety of the distal–proximal acupoints combination remain to be confirmed. Additionally, which form of acupoints combination is better is also worth studying in the future.

## Mechanism of acupuncture

Intestinal manipulation in GI surgery causes autonomic dysfunction, inflammatory activation, agonism at intestinal opioid receptors, modulation of GI hormone activity, and electrolyte derangements [3]. These events lead to significant delays in GI transit and finally result in POI. Acupuncture can directly induce motility acceleration to restore GI transit through the parasympathetic efferent pathway [54, 55]. The main factor responsible for the prolonged dysmotility of the GI tract associated with POI is intestinal inflammation [56]. The recent studies by Yang et al. [57, 58] showed that EA can alleviate intestinal inflammation via activation of the α7nAChR-mediated Janus kinase 2/signal transducer and activator of transcription 3 (JAK2/STAT3) signaling pathway in POI. In addition, several molecules involved in the inflammation, such as nitric oxide (NO), have a direct effect on intestinal contractility. After intestinal manipulation, NO disrupts the generation and propagation of pacemaker potentials by interstitial cells of Cajal (ICC). Deng et al. [59, 60] found that acupuncture can improve postoperative GI motility by facilitating ICC recovery. The potential mechanism may illustrate our findings that acupuncture promotes postoperative intestinal function recovery and reduces POI.

## Limitations

There were some unavoidable limitations in this review. First, the population that underwent GI surgery in the included trials was mostly Asians. Therefore, our evidence should be used prudently in other regions and other surgeries. Second, significant heterogeneity was observed when investigating the effect of acupuncture. However, the subgroup analysis did not address the heterogeneity. Considering that GI surgery is highly complex, multiple factors—such as the type of surgical approaches, usual care mode, and anesthesia method—may account for heterogeneity. Third, the safety of acupuncture was not fully evaluated due to the limited number of trials. Last, based on the fact that the purpose of the present study was to assess the effectiveness and safety of acupuncture for POI following GI surgery, we considered only RCTs for inclusion in this review.

RCTs are considered the gold standard of evidence-based medicine for health interventions because they are designed to minimize the risk of bias. A non-randomized controlled trial

(NRCT) could not satisfactorily eliminate possible biases due to other factors (apart from treatment), which may affected the results by other confounding factors and may severely compromise the validity of their results [61]. However, the applicability of RCT results is limited due to restrictive selection criteria. In contrast, NRCTs are generally more likely to reflect real-life clinical practice because they have a wider range of participants and longer follow-up. Incorporating data from NRCTs to complement RCTs can generate more comprehensive evidence to guide healthcare decisions [62, 63]. In our future research, we will include data from NRCTs to assess acupuncture from a more pragmatical perspective.

### Implications for research

There is a need for large, high-quality, multicenter RCTs to further determine the effectiveness and safety of acupuncture in populations beyond the Asian area. Larger studies may also help identify the clinical difference in GI surgery details.

Given the discrepant results for acupoint combination and frequency of treatment sessions in the subgroup analysis, future studies could focus specifically on the acupuncture therapeutic parameters of acupoints combination, frequency, stimulation and duration to formulate optimal acupuncture treatment scheme for POI following GI surgery. In addition, to better evaluate the safety of acupuncture, the description of acupuncture operation details and adverse events should be clearly reported in future research according to the Standards for Reporting Interventions in Clinical Trials of Acupuncture (STRICTA) guideline [64].

Increased health care costs are associated with prolonged hospital stay in POI patients [9]. Previous study has shown that 1-day earlier hospital time to discharge contributed to potentially beneficial for overall healthcare costs [65]. In this meta-analysis, acupuncture had an advantage in reducing LOS compared with usual care by an average of 1.68 days. It is worth to address whether acupuncture may bring potential economic benefits by reducing LOS in future studies.

### Conclusions

In conclusion, the evidence of this systematic review showed that acupuncture has certain effect in reducing TFF, TFD, TBSR and LOS compared to usual care. Acupuncture may be considered as a promising intervention in the management of POI following GI surgery. Taking into consideration of the very low to moderate quality of the overall body of evidence, the findings from this systematic review should be interpreted with caution. High-quality, large-sample, multi-center original studies are needed in the future.

### Supporting information

**S1 Checklist. PRISMA 2020 checklist.**
(DOCX)

**S1 Appendix. Search strategy.**
(DOCX)

**S2 Appendix. Sensitivity analysis.**
(DOCX)

### Author Contributions

**Conceptualization:** Zi Ye, Ke Wang, Jia Zhou.

**Data curation:** Zi Ye, Shouquan Feng, Qunhao Gu.

**Formal analysis:** Zi Ye, Jing Li.

**Funding acquisition:** Ke Wang.

**Investigation:** Zi Ye, Shouquan Feng, Qunhao Gu.

**Methodology:** Zi Ye, Xuqiang Wei, Le Kuai.

**Project administration:** Ke Wang, Jia Zhou.

**Resources:** Ke Wang, Jia Zhou.

**Software:** Zi Ye, Yue Luo, Ziqi Xi.

**Supervision:** Zi Ye, Xuqiang Wei, Ke Wang, Jia Zhou.

**Validation:** Zi Ye, Jing Li, Le Kuai, Ke Wang, Jia Zhou.

**Visualization:** Zi Ye, Yue Luo, Ziqi Xi.

**Writing – original draft:** Zi Ye.

**Writing – review & editing:** Zi Ye, Xuqiang Wei, Ke Wang, Jia Zhou.

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
