## [Decision Letter · Decision Letter 0]

18 Mar 2022

PONE-D-22-01716Effectiveness and safety of acupuncture for postoperative ileus following gastrointestinal surgery: A systematic review and meta-analysisPLOS ONE

Dear Dr. Wang,

Thank you for submitting your manuscript to PLOS ONE. After careful consideration, we feel that it has merit but does not fully meet PLOS ONE’s publication criteria as it currently stands. Therefore, we invite you to submit a revised version of the manuscript that addresses the points raised during the review process.

Based on PLOS ONE’s publication criteria this manuscript can be improved according to suggestions of reviewers reported below. 

We look forward to receiving your revised manuscript.

Kind regards,

Sandro Pasquali, M.D., Ph.D.

Academic Editor

PLOS ONE

Journal Requirements:

"This work was supported by the 2019 Project of Building Evidence Based Practice Capacity for TCM [2019XZZX-ZJ0011]; the Shanghai Clinical Research Center for Acupuncture and Moxibustion Accelerating [20MC1920500]; the Development of Chinese Medicine Three-Year Action Plan of Shanghai [ZY (2018-2020)-CCCX-2004-04]; the Scientific Research Project of Shanghai Municipal Health Commission [201840011]; the Clinical Key Specialty Construction Foundation of Shanghai [shslczdzk04701]; and the Shanghai Health System Talent Training Program [2018BR24]. "

Reviewers' comments:

Reviewer's Responses to Questions

**Comments to the Author**

1. Is the manuscript technically sound, and do the data support the conclusions?

Reviewer #1: Yes

Reviewer #2: Yes

Reviewer #3: Yes

2. Has the statistical analysis been performed appropriately and rigorously? 

Reviewer #1: Yes

Reviewer #2: Yes

Reviewer #3: Yes

3. Have the authors made all data underlying the findings in their manuscript fully available?

Reviewer #1: Yes

Reviewer #2: Yes

Reviewer #3: Yes

4. Is the manuscript presented in an intelligible fashion and written in standard English?

Reviewer #1: Yes

Reviewer #2: Yes

Reviewer #3: Yes

5. Review Comments to the Author

Reviewer #1: Thank you for the opportunity to review this manuscript – I have found the topic interesting and important to the field of gastrointestinal surgery. A systematic review and meta-analysis of the impact of acupuncture for post-operative ileus (POI).

The authors tackle a major challenge in gastrointestinal surgery. The systematic review and meta-analysis are rigorous and adhere to recognised checklists for this research type. The authors correctly state the current literature, involving several meta-analysis with conflicting evidence. The paper presented is an update to existing literature from recent years, limiting the significance of its findings.

Please consider the following comments.

1. The authors describe their methodology in detail and outline the inclusion criteria. Please could the authors state why only RCTs were included and non-randomised trials were not considered for inclusion.

2. The methods also describe the search strategy, please could the authors state if expert support was sought to define the search strategy, such as from an information specialist.

3. The authors correctly investigate risk of bias across all studies. Several RCTs included are reported to have high risk of bias components or unclear risk of bias. Please could the authors further define how this was accounted for in the meta-analysis.

Reviewer #2: The article describes an important and actual gastrointenstinal complication, dealing with the surgical manipulation of the bowel and tries to highlight the role of the acupuncture as an integrative therapy to treat and reduce this condition. Title and abstract are appropriate for the content of the article: the abstract summarize accurately the key-points.

It’s well known that gastrointenstinal surgery can induce a transient paralysis of intestinal motility, called POI (post-operative ileus), which is accompanied by abdominal symptoms, general discomfort of the patient, long hospitalization and increased costs. Etiology of POI is multifactorial and its related to surgery but also to the use of opioid analgesia, perioperative stress with the necessity of anxiolitic drugs, prolonged supine position with delayed ambulation.

The approach to this complication is multidisciplinary and this is the aim of the article, suggesting the use of the acupuncture as a complementary approach. Acupunture is a very ancient technique of the Traditional Chinese Medicine; its role is to balance the flow of energy of the body, involving the insertion of thin needles through the patient’s skin in specific acupoints. Acupuncture is being used to relieve discomfort associated with a variety of diseases and conditions (chemotherapy-induced and postoperative nausea and vomiting; pain; stress and mood disorders; gynecologic, neurological, respiratory and gastrointestinal disorders; for the overall wellness). The benefits of acupuncture are sometimes difficult to measure, so in this article its efficacy is related to time to first flatus (TFF) and time to first defecation (TFD) as primary outcomes, and to time to bowel sounds recovery (TBSR) and length of hospital stay (LOS) as secondary outcomes compared to usual care. In my opinion this article lacks of explanation of what really means usual care in this particular clinical condition. There is only an hint in the “Introduction” (lines 76-78) of potential treatments to reduce the occurrence of POI.

However, data extraction process is well performed; figures, tables, data presentation and statistical analysis are clear to consult, complete and adeguate.

In the paragraph “Acupoints combination” are mentioned two of the acupuncture’s basic method to treat, the combination of distal and proximal acupoints, but it’s not well explained what does it mean (does proximal mean closer to the gastro-intestinal district?) I think it’s important and more complete to point out the difference. The authors mention the prevalence of the use of ST36, but this is not explained why. ST 36 is one of the master point in Traditional Chinese Medicine, due to its efficacy and tropism for all gastro-intestinal disorders, that’s the reason why is so largely used.

The authors can support the conclusions with the presented results, even if it’s clearly pointed out the limitation of the reviewed studies and the necessity of more studies.

In conclusion, in my opinion this review is a valid consultation tool for the acupuncturist who wants to deepen the topic and orient the therapeutic technique.

Reviewer #3: This systematic review and meta-analysis wanted to assess the effectiveness and safety of acupuncture for postoperative ileus in patients after gastrointestinal surgery. The report of this review followed the PRISMA statement and remained to be reported more clearly.

1.It proposed that the Table 1 should report more details, including course of treatment and a brief information of usual care in the control group.

2.For outcomes time to first flatus, time to first defecation and time to bowel sounds recovery, SMD was used as an effect size to synthesize data, which does not contribute to the understanding of the clinical importance of the difference. What is the rationale?

3.It suggests adding the description for any methods used to assess certainty in the body of evidence for some outcomes and their results.

6. PLOS authors have the option to publish the peer review history of their article (what does this mean?). If published, this will include your full peer review and any attached files.

Reviewer #1: No

Reviewer #2: No

Reviewer #3: No

---

## [Author Response · Author response to Decision Letter 0]

11 May 2022

Reviewer #1:

1. Comment: The authors describe their methodology in detail and outline the inclusion criteria. Please could the authors state why only RCTs were included and non-randomised trials were not considered for inclusion.

Answer: Thank you for your comment. RCTs are considered the gold standard of evidence-based medicine for health interventions because they are designed to minimize the risk of bias. The non-randomised trials (NRSI) could not satisfactorily eliminate possible biases due to other factors (apart from treatment), which results may be affected by other confounding factors and may severely compromise the validity of their results [Altman DG, Bland JM. Statistics notes. Treatment allocation in controlled trials: why randomise? BMJ. 1999, 318(7192):1209.]. Randomized design can balance potential confounding factors and ensure the homogeneity of the two groups of patients. Therefore, evidence in RCT studies has high internal consistency and can better reflect the net effect of acupuncture therapies. Furthermore, there is no difference on average in the risk estimate of adverse effects of an intervention derived from meta-analyses of RCTs and meta-analyses of observational studies [Golder S, Loke YK, Bland M. Meta-analyses of adverse effects data derived from randomised controlled trials as compared to observational studies: methodological overview. PLoS Med. 2011, 8(5):e1001026.]. Since the purpose of the present study was to assess the effectiveness and safety of acupuncture for postoperative ileus following gastrointestinal surgery, we considered only RCTs for inclusion in this review. 

We also noticed that the applicability of RCT results is limited due to restrictive selection criteria. In contrast, NRSI are generally more likely to reflect real-life clinical practice because they have a wider range of participants, longer follow-up time. Incorporate data from NRSIs to complement RCTs can generate more comprehensive evidence to guide healthcare decisions [1. Faber T, Ravaud P, Riveros C, Perrodeau E, Dechartres A. Meta-analyses including non-randomized studies of therapeutic interventions: a methodological review. BMC Med Res Methodol. 2016, 22;16:35. 2. Sarri G, Patorno E, Yuan H, Guo JJ, Bennett D, Wen X, Zullo AR, Largent J, Panaccio M, Gokhale M, Moga DC, Ali MS, Debray TPA. Framework for the synthesis of non-randomised studies and randomised controlled trials: a guidance on conducting a systematic review and meta-analysis for healthcare decision making. BMJ Evid Based Med. 2022, 27(2):109-119.]. This limitation and recommendation have been strengthened and pointed out in revised manuscript. After a systematic review of the evidence from the RCTs, we will review the data from NRSIs from a more pragmatical perspective in the future.

2. Comment: The methods also describe the search strategy, please could the authors state if expert support was sought to define the search strategy, such as from an information specialist.

Answer: Yes. Our work is a part of the project funded by Shanghai Clinical Research Center for Acupuncture and Moxibustion (No. 20MC1920500) and National Administration of Traditional Chinese Medicine: 2019 Project of building evidence based practice capacity for TCM (No. 2019XZZX-ZJ0011). Therefore, the search strategy of this work was guided by the expert group of Shanghai Clinical Research Center for Acupuncture and Moxibustion and China Center for Evidence Based Traditional Chinese Medicine--Acupuncture and Moxibustion Advantageous Diseases Project Team, which included information specialist.

3. Comment: The authors correctly investigate risk of bias across all studies. Several RCTs included are reported to have high risk of bias components or unclear risk of bias. Please could the authors further define how this was accounted for in the meta-analysis.

Answer: Thank you for your rigorous consideration. According to your suggestion, we have added assessment details in the Result section and judgements about the risk of bias affecting the quality of evidence in the discussion section to clarify this point.

Risk-of-bias assessment is a central component of systematic reviews. We assessed the methodological qualities of all studies in strict accordance with the Cochrane handbook [Higgins JPT, Altman DG, Sterne JAC (editors). Chapter 8: Assessing risk of bias in included studies. In: Higgins JPT, Churchill R, Chandler J, Cumpston MS (editors), Cochrane Handbook for Systematic Reviews of Interventions version 5.2.0 (updated June 2017), Cochrane, 2017.]. In Cochrane handbook, the criteria for the judgement of ‘high risk’ of bias in this domain was: no blinding or incomplete blinding, and the outcome was likely to be influenced by lack of blinding; blinding of key study participants and personnel attempted, but likely that the blinding could have been broken, and the outcome was likely to be influenced by lack of blinding. In this study, we found that some trials didn’t use blind method so we judged them as ‘high risk’. The criteria for the judgement of ‘unclear risk’ of bias in the domain of allocation concealment was: Insufficient information available to permit a judgement of ‘low risk’ or ‘high risk’ which usually the case if the method of concealment is not described or not described in sufficient detail to allow a definite judgement – for example if the use of assignment envelopes was described, but it remains unclear whether envelopes were sequentially numbered, opaque and sealed. In this study, there were thirteen trials didn’t report the details of the allocation concealment so that we permit a judgment of ‘unclear risk’. The criteria for the judgement of ‘unclear risk’ of bias in the domain of blinding of outcome assessment was: insufficient information available to permit a judgement of ‘low risk’ or ‘high risk’; the study did not address this outcome. Fifteen included trials in this meta-analysis didn’t adequately descripted whether the outcome assessors were blinded to the treatment allocation so were judged as ‘unclear risk’.

The results of the assessment showed that the high risk of bias mainly existed in the domain of ‘blinding of participants and personnel’, and the unclear risk mainly existed in the domain of ‘allocation concealment’ and ‘blinding of outcome assessment’. Serious risk of bias was the main reason for rating down the quality of evidence. The lack of blinding and uncertain in allocation concealment may lead to an overestimation or underestimation of the true effects of intervention [Reveiz L, Chapman E, Asial S, Munoz S, Bonfill X, Alonso-Coello P. Risk of bias of randomized trials over time. J Clin Epidemiol. 2015; 68(9):1036–45.]. However, it should be noted that blinding of participants and practitioners is difficult due to the nature of acupuncture intervention. But as far as possible the participants should be blinded whenever possible to ensure that participants cannot distinguish the real acupuncture from a sham control. In addition, the blinding of outcome assessors and statisticians is necessary and the specific process should be reported.

Reviewer #2:

1. Comment: In my opinion this article lacks of explanation of what really means usual care in this particular clinical condition. There is only an hint in the “Introduction” (lines 76-78) of potential treatments to reduce the occurrence of POI

Answer: Thank you for your suggestion. We have added definitions of usual care to the introduction section. The usual care patient received after GI surgery mainly includes routine nasogastric tubes, intravenous fluids, parenteral nutrition and early mobilization [Wattchow D, Heitmann P, Smolilo D, Spencer NJ, Parker D, Hibberd T, Brookes SSJ, Dinning PG, Costa M. Postoperative ileus-An ongoing conundrum. Neurogastroenterol Motil. 2021 May;33(5):e14046.]. Meanwhile, according to reviewer #3’s suggestion, we also supplemented a brief information of usual care in each included RCT in the Table 1.

2. Comment: In the paragraph “Acupoints combination” are mentioned two of the acupuncture’s basic method to treat, the combination of distal and proximal acupoints, but it’s not well explained what does it mean (does proximal mean closer to the gastro-intestinal district?) I think it’s important and more complete to point out the difference.

Answer: We are very sorry we didn't make it clear, your understanding is correct. We have added definitions to the discussion of “Acupoints combination” section.

3. Comment: The authors mention the prevalence of the use of ST36, but this is not explained why. ST 36 is one of the master point in Traditional Chinese Medicine, due to its efficacy and tropism for all gastro-intestinal disorders, that’s the reason why is so largely used.

Answer: Thank you for your important suggestion. We have added this point into the discussion of “Acupoints combination” section. Thanks again.

Reviewer #3:

1. Comment: It proposed that the Table 1 should report more details, including course of treatment and a brief information of usual care in the control group.

Answer: Thank you for the valuable suggestions. We have supplement details of the course of treatment and brief information of usual care in Table 1. 

2. Comment: For outcomes time to first flatus, time to first defecation and time to bowel sounds recovery, SMD was used as an effect size to synthesize data, which does not contribute to the understanding of the clinical importance of the difference. What is the rationale?

Answer: Thank you for pointing out an important question. The mean difference (MD) is the difference in the means of the treatment group and the control group, while the standardized mean difference (SMD) is the MD divided by the standard deviation (SD), derived from either or both of the groups. As you said, the overall treatment effect in terms of SMD can be difficult to interpret as it is reported in units of standard deviation rather than in units of any of the measurement scales used in review [Egger M, Smith GD, Altman D. Systematic Reviews in Health Care: Meta-Analysis in Context. 2008. Wiley. com.]. However, the MD from different RCTs with different units cannot be pooled in meta-analysis to yield a summary estimate. Part 2, Chapter 9, of the Cochrane Handbook pointed out that the choice between MD and SMD depends on whether “outcome measurements in all studies are made on the same scale.” [Higgins JP, Green S. Cochrane Handbook for Systematic Reviews of Interventions, Version 52.0 [Updated June 2017]. London, UK: The Cochrane Collaboration; 2017]. One study suggested that the SMD does not depend on the unit of measurement, and therefore the SMD has been widely used as a measure of intervention effect in many applied fields [Tian L. Inferences on standardized mean difference: the generalized variable approach. Stat Med. 2007 Feb 28;26(5):945-53.]. The use of SMD in this study is based on the following considerations: First, the time to first flatus, time to first defecation and time to bowel sounds recovery are three important evaluation indicators to measure the clinical efficacy of acupuncture for POI following GI surgery. The units of measurement for these outcomes are different across the included studies, some are measured in days, some are measured in hours. Second, Although the changes of SMD couldn’t directly reflect the clinical importance of the difference since it doesn’t have units, we can use the Cohen’s d as the measure of effect size to show the significance of difference between two groups [Andrade C. Mean Difference, Standardized Mean Difference (SMD) and Their Use in Meta-Analysis: As Simple as It Gets. J Clin Psychiatry. 2020 Sep 22;81(5):20f13681.]. It is widely accepted that SMD values of 0.2-0.5 are considered a “small” effect, values of 0.5-0.8 are considered a “medium” effect, and values > 0.8 are considered a “large” effect [Patrick Schober. Statistics From A (Agreement) to Z (z Score): A Guide to Interpreting Common Measures of Association, Agreement, Diagnostic Accuracy, Effect Size, Heterogeneity, and Reliability in Medical Research; Cohen J. The t test for means. In: Statistical Power Analysis for the Behavioral Sciences. Psychology Press, Taylor & Francis Group, 1988:19–74.]. Third, studies for which the difference in means is the same proportion of the standard deviation (SD) will have the same SMD, regardless of the actual scales used to make the measurements [Cochrane Handbook version 5.2.0 Chapter 6.5.1.2]. So the SMD be likewise concerned as MD because this means that the curve representing the distribution has been shifted to the right by the same whole SD [Andrade C. Mean Difference, Standardized Mean Difference (SMD) and Their Use in Meta-Analysis: As Simple as It Gets. J Clin Psychiatry. 2020 Sep 22;81(5):20f13681.]. Forth, a study, which included 1068 meta-analyses, found that there were no differences in the percentage of statistical significance between MD and SMD in either model [Takeshima N, Sozu T, Tajika A, Ogawa Y, Hayasaka Y, Furukawa TA. Which is more generalizable, powerful and interpretable in meta-analyses, mean difference or standardized mean difference? BMC Med Res Methodol. 2014 Feb 21;14:30.]. Therefore, we use SMD to combine the outcomes in the meta-analyses. We have added text and clarification in the revised manuscript.

3. Comment: It suggests adding the description for any methods used to assess certainty in the body of evidence for some outcomes and their results.

Answer: We appreciate the reviewer’s helpful suggestion. In the revised manuscript, we have added the description of GRADE approach which assessed the level of certainty in the evidence for intervention to the Method section. In addition, we also supplement the summary of certainty assessment results to the Results section and discussed the quality of evidence.

---

## [Decision Letter · Decision Letter 1]

15 Jun 2022

PONE-D-22-01716R1Effectiveness and safety of acupuncture for postoperative ileus following gastrointestinal surgery: A systematic review and meta-analysisPLOS ONE

Dear Dr. Wang,

Thank you for submitting your manuscript to PLOS ONE. After careful consideration, we feel that it has merit but does not fully meet PLOS ONE’s publication criteria as it currently stands. Therefore, we invite you to submit a revised version of the manuscript that addresses the points raised during the review process.

A revision of English writing is needed before acceptance. 

We look forward to receiving your revised manuscript.

Kind regards,

Sandro Pasquali, M.D., Ph.D.

Academic Editor

PLOS ONE

Journal Requirements:

Additional Editor Comments:

Authors did reply thoroughly to reviewers' comments. Before the manuscript can be accepted a revision of English writing is required, for instance contracted forms should be avoided. Please resubmit a proof-read version of this article.

Reviewers' comments:

Reviewer's Responses to Questions

**Comments to the Author**

1. If the authors have adequately addressed your comments raised in a previous round of review and you feel that this manuscript is now acceptable for publication, you may indicate that here to bypass the “Comments to the Author” section, enter your conflict of interest statement in the “Confidential to Editor” section, and submit your "Accept" recommendation.

Reviewer #2: All comments have been addressed

Reviewer #3: All comments have been addressed

2. Is the manuscript technically sound, and do the data support the conclusions?

Reviewer #2: Yes

Reviewer #3: Yes

3. Has the statistical analysis been performed appropriately and rigorously? 

Reviewer #2: Yes

Reviewer #3: Yes

4. Have the authors made all data underlying the findings in their manuscript fully available?

Reviewer #2: Yes

Reviewer #3: Yes

5. Is the manuscript presented in an intelligible fashion and written in standard English?

Reviewer #2: Yes

Reviewer #3: Yes

6. Review Comments to the Author

Reviewer #2: (No Response)

Reviewer #3: The authors have responded appropriately to my comments. The revised manuscript sounds that followed the PRISMA statement well.

7. PLOS authors have the option to publish the peer review history of their article (what does this mean?). If published, this will include your full peer review and any attached files.

Reviewer #2: No

Reviewer #3: No

---

## [Author Response · Author response to Decision Letter 1]

25 Jun 2022

Dear Editor:

Thank you for arranging a timely review for our manuscript entitled “Effectiveness and safety of acupuncture for postoperative ileus following gastrointestinal surgery: A systematic review and meta-analysis” (PONE-D-22-01716R1). 

The manuscript has been polished by an English language editing company (https://www.cwauthors.com/). We revised the manuscript and marked all changes in yellow. We also reviewed reference list and ensured that they were complete and correct. The cited papers were not retracted.

---

## [Editor Report · Decision Letter 2]

4 Jul 2022

Effectiveness and safety of acupuncture for postoperative ileus following gastrointestinal surgery: A systematic review and meta-analysis

PONE-D-22-01716R2

Dear Dr. Wang,

We’re pleased to inform you that your manuscript has been judged scientifically suitable for publication and will be formally accepted for publication once it meets all outstanding technical requirements.

Kind regards,

Sandro Pasquali, M.D., Ph.D.

Academic Editor

PLOS ONE
---

## [Editor Report · Acceptance letter]

8 Jul 2022

PONE-D-22-01716R2 

Effectiveness and safety of acupuncture for postoperative ileus following gastrointestinal surgery: A systematic review and meta-analysis 

Dear Dr. Wang:

I'm pleased to inform you that your manuscript has been deemed suitable for publication in PLOS ONE. Congratulations! Your manuscript is now with our production department. 

Kind regards, 

on behalf of

Dr. Sandro Pasquali 

Academic Editor

PLOS ONE